# Learning Differential Pyramid Representation for Tone Mapping

## Abstract

To display high dynamic range (HDR) images on low dynamic range (LDR) screens, tone mapping operations (TMO) are required to compress the dynamic range. Recently, deep learning-based TMO methods have shown promising performance. However, these methods often fail to deliver satisfactory results in local areas, and generating image-level TMO on down-sampled low-resolution images leads to loss of details. To address this issue, we propose a learnable Differential Pyramid Representation Network (DPRNet), a comprehensive framework capable of recovering details and manipulating global and local tone mappings. Specifically, we construct a Global Tone Perception module to modulate the image globally. Then the Local Tone Tuning generates TMO coefficients at the patch level to refine the local tones. Furthermore, we propose a Learnable Differential Pyramid module to capture multi-scale high-frequency components, coupled with an Iterative Detail Enhancement progressively refining image resolution and image details. Extensive experiments demonstrate that our method significantly outperforms state-of-the-art methods, improving PSNR by 2.58 dB in the HDR+ dataset and 3.31 dB in the HDRI Haven dataset respectively compared with the second-best method. In addition, our method has the best generalization ability in the unsupervised image and video TMO. *We provide an anonymous online demo at* [https://xxxxxx2024.github.io/DPRNet/](https://xxxxxx2024.github.io/DPRNet/).

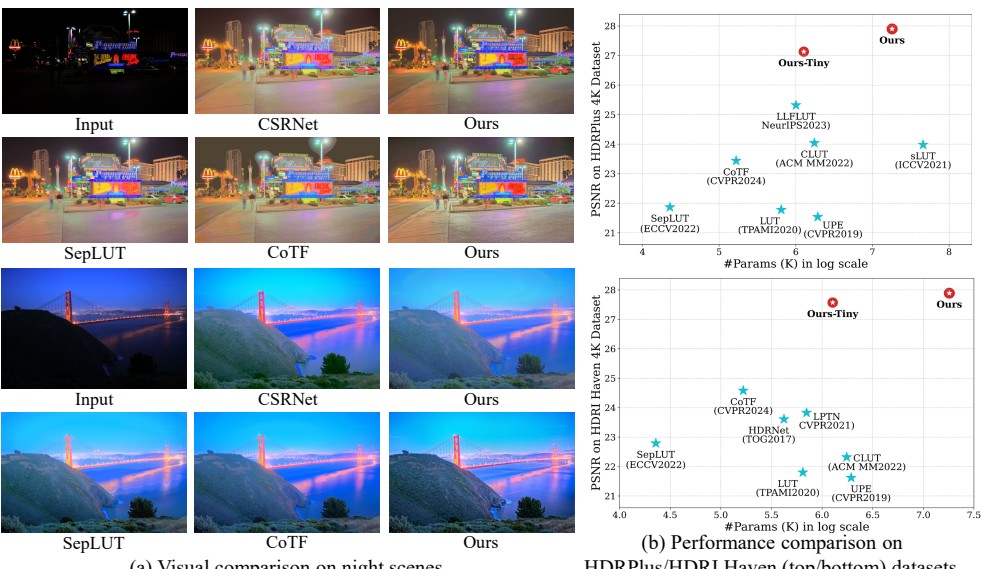

(a) Visual comparison on night scenes.

(b) Performance comparison on HDRPlus/HDRI Haven (top/bottom) datasets.

Figure 1: Motivation and superiority. (a) Existing global tone mapping methods lead to the loss of details and dead blacks. (b) Our DPRNet provides remarkable performance gains on the HDRplus and HDRI Haven datasets.

## 1 Introduction

High dynamic range imaging has revolutionized the way we capture and represent real-world scenes, preserving rich luminance details across varied intensity ranges. However, the challenge of effectively rendering these HDR images on conventional low dynamic range (LDR) display devices remains due to inherent hardware constraints. In this context, tone mapping (TM) algorithms emerge as a critical bridge, narrowing the dynamic range disparity between HDR content and LDR screens.

Recent methods based on deep neural networks (DNNs) [1, 2, 3] have made impressive progress in tone mapping performance. These methods [4, 5, 6, 7] can be broadly classified into two categories: global and local tone mapping, based on their operational patterns. Global tone mapping [2, 8] focuses on learning the dense pixel-wise mapping or transformation between input HDR image and output LDR image pairs, including pixel-level and image-level tone mapping. These methods, without considering neighboring pixels, ignore spatial transformations and intensity correlations between pixels, resulting in over-compression of bright areas or loss of detail in dark areas, see Fig. 1. Conversely, local tone mapping [9, 10], patch-level TMO, can dynamically adjust the tones of local regions based on local luminance features. Despite their local adaptivity, these methods suffer from halo artifacts and exhibit substantial computational demands, particularly for high-resolution images.

On the other hand, we observe that the customized software, e.g., photoshop and Photomatix, often includes both global and local tone mapping operations [11, 12], where the global TMO is realized by logarithmic compression, and the local TMO is realized by local laplacian pyramids. However, these methods lack flexibility in dealing with various image contents. *Therefore, integrating local and global TMO in a unified learning framework is expected.*

Furthermore, to reduce computational resources, existing methods [5, 7, 13] usually manipulate tone mapping at a low resolution after image downsampling. This method introduces a trade-off, where lower resolution processing inevitably leads to a compromise in image detail and quality. To address this problem, some methods [4, 14] have introduced reversible pyramid decomposition [15] to maintain and recover high-frequency details through multi-scale decomposition. *However, the extraction of high-frequency components from HDR images is inherently challenging due to their extensive dynamic range and complex texture structure.* The strong contrast and rich lighting variations of HDR images make it difficult for traditional manual features to effectively capture these details [4, 7]. Especially in extremely bright or dark areas, image details are often compressed or lost. Fig. 1 illustrate that the enhancement results of the existing methods suffer from severe loss of detail.

To address these challenges, we propose a learnable Differential Pyramid Representation Network (DPRNet), a comprehensive framework designed to recover details and manipulate global and local tone mappings. Our method involves the decomposition of tone mapping into three distinct yet interrelated subtasks: i) global tone perception, which manipulates the overall tonality and preserves image visual consistency, ii) local tone tuning, which refines the brightness and contrast of specific regions in a patch-based manner; iii) high-frequency extraction and reconstruction, which restores image resolution while enhancing detail fidelity. To achieve comprehensive tone mapping, we design a Global Tone Perception module for the global modulation of image and then construct a Local Tone Tuning module to refine the local tone. To compensate for detail loss during downsampling, we propose a Learnable Differential Pyramid module to capture multi-scale high-frequency components, coupled with an Iterative Detail Enhancement strategy that facilitates a progressive enhancement of detail while restoring image resolution. This framework enables the refinement of each level of the pyramid images to produce multi-scale LDR images for supervision, thereby enabling the network to learn the mapping between HDR and LDR images across multiple scales.

Our contributions are summarized as follows:

- We propose an innovative end-to-end network, the Differential Pyramid Representation Network (DPRNet), capable of detail recovery and the simultaneous manipulation of global and local tone mappings.

- We introduce the Learnable Differential Pyramid module for adaptive high-frequency extraction, coupled with an Iterative Detail Enhancement module to progressively restore image resolution and detail.

- Extensive experiments demonstrate that our method significantly outperforms state-of-the-art methods, improving PSNR by 2.58 dB in the HDR+ dataset and 3.31 dB in the HDRI Haven dataset respectively compared with the second-best method. In addition, our method has the best generalization ability in the unsupervised image and video TMO.

## 2 RELATED WORKS

### 2.1 LEARNING-BASED TONE MAPPING METHODS

Recent advancements in tone mapping have leveraged deep learning techniques to address the nonlinear mapping from HDR to LDR images. Hou et al. [2] applied CNNs to tone mapping tasks,

establishing a foundation for subsequent research. He et al. [8] developed a conditional sequential retouching network for effective image retouching. Similarly, Cao et al. [1], Rana et al. [16], and Panetta et al. [17] explored generative adversarial networks (GANs) for pixel-precise mapping. Despite these efforts, issues such as halo effects, noise, and local area processing remain challenging. To combat this, Hu et al. [18] combined tone mapping with denoising, incorporating a discrete cosine transform module for noise removal and improved image quality. Zhang et al. [19] manipulated tone in the HSV color space, which significantly reduced halos and preserved detail. Although these learning-based methods have achieved remarkable results, most of them are either global or local mappings, struggling to achieve satisfactory tone mapping results.

## 2.2 3D LUT-BASED ENHANCEMENT METHODS

In pursuit of computational efficiency and performance, emerging tone mapping methods employed 3D LUT to render image brightness and color. Zeng et al. [5] developed an adaptive 3D LUTs prediction network that merges multiple foundational 3D LUTs. Zhang et al. [13] presented a compressed representation of 3D LUTs, reducing parameters. Wang et al. [20] studied pixel-level fusion based on 3D LUTs, extending the method to a spatially aware variant. Yang et al. [21] proposed effective ICELUT for extremely efficient edge inference. Yang et al. [22] introduced the AdaInt mechanism, which adaptively learns non-uniform sampling intervals in the 3D color space for more flexible sampling point allocation. Furthermore, Yang et al. [7] proposed SepLUT, addressing the shortcomings of 1D LUTs in color component interaction while alleviating the memory consumption issues of 3D LUTs. Despite these efforts, these methods necessitate an initial down-sampling step to reduce the network computation, and in the case of 4K images, even need 16 times down-sampling. As a result, this leads to loss of image details and degradation of image quality. Zhang et al. [4] attempted to use the Laplacian pyramid to compensate for the loss of high-frequency detail, but effectively capturing the high-frequency component of HDR images remains a challenge.

## 3 METHODS

### 3.1 MOTIVATION

Combining the strengths of both methods achieves a critical balance: global tone mapping ensures overall consistency in luminance, while local tone mapping preserves fine detail and improves local contrast in challenging lighting conditions. As a result, existing customized software often includes both global and local tone mapping operations. However, few works have integrated global and local tone mapping strategies into a unified learning framework. Comprehensive tone mapping not only solves the limitations of global tone mapping, i.e., enhancing detail retention in complex lighting scenes but also mitigates the problems of localized tone mapping, such as artifacts and inconsistencies. On the other hand, paradigms based on frequency domain decomposition have been widely used for various image-processing tasks [4, 23, 24]. Previous methods typically utilized pyramids [15] with fixed convolutional kernels to manually extract features. However, limited by complex lighting environments, traditional manual features often struggle to capture high-frequency details effectively. Therefore, adaptively extracting high-frequency details is a critical step in tone mapping.

### 3.2 FRAMEWORK OVERVIEW

To handle global and local tone mapping while preserving high-frequency details, we propose an end-to-end framework, the Differential Pyramid Representation Network (DPRNet). The pipeline of the proposed DPRNet is shown in Fig. 2. Our framework is composed of four components: global tone perception, local tone tuning, learnable differential pyramid, and iterative detail enhancement. Given an HDR image $\mathbf{X} \in \mathbb{R}^{H \times W \times 3}$, our aims to generate an LDR image $\mathbf{T}_0 \in \mathbb{R}^{H \times W \times 3}$. We begin to manipulate tone mapping from downsampled low-resolution images $\mathbf{L}_n$, optimizing for computational complexity. Subsequently, we employ the Global Tone Perception (GTP) to manipulate the global tone and preserve image visual consistency. To refine the local regions, we built the Local Tone Tuning (LTT) module to refine the brightness and contrast of different regions in a patch manner. The combination of GTP and LTT allows our framework to maintain the overall tone of an image while adaptively enhancing local contrast and details. To compensate for the loss of detail due to downsampling, we introduce a Learnable Differential Pyramid for adaptively extracting high-frequency information from complex scenes. Finally, we utilize the high-frequency details captured by the LDP to refine the texture layer by layer while restoring the resolution through iterative detail enhancement modules.

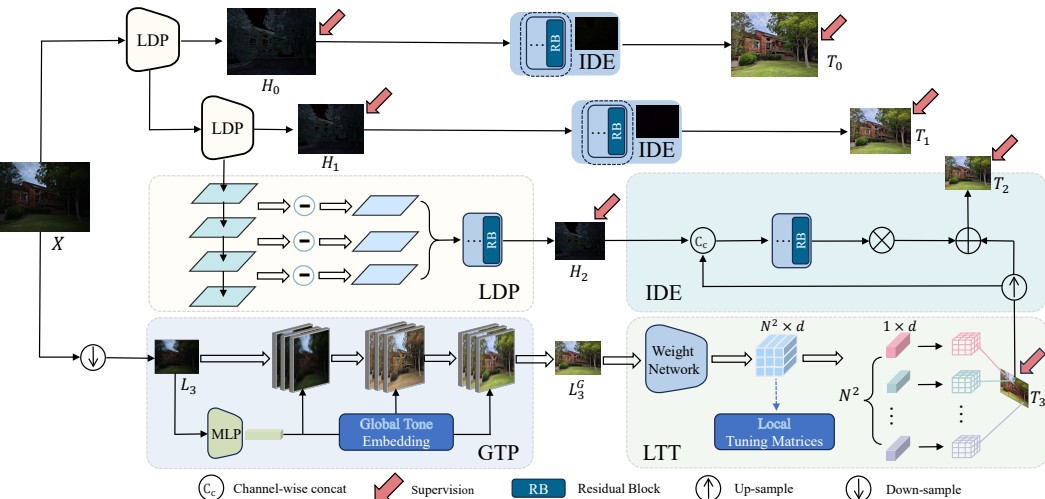

Figure 2: Overview of the proposed DPRNet. Our DPRNet contains three main components: 1) a Global Tone Perception (GTP) module that handles the global tone, 2) a Local Tone Tuning (LTT) module that adjusts brightness and color in a patch manner, and 3) Learnable Differential Pyramid (LDP) modules and Iterative Detail Enhancement (IDE) module extract high-frequency information and restore image details.

### 3.3 GLOBAL TONE PERCEPTION

Global tone mapping typically applies a uniform mapping function to manipulate global brightness and contrast. We mathematically find that these pixel-independent operations can be approximated or formulated by multi-layer perceptrons (MLPs). For global brightness adjustment, given an input image $\mathbf{X}$, the global brightness is described as the average value of its luminance map:

$$\mathbf{X}_Y = 0.299 * \mathbf{X}_R + 0.578 * \mathbf{X}_G + 0.114 * \mathbf{X}_B \tag{1}$$

where $\mathbf{X}_R$, $\mathbf{X}_G$, $\mathbf{X}_B$ represent the R/G/B channels, respectively. One simple way to adjust the brightness is to multiply a scalar for each pixel:

$$\mathbf{X}'_Y(x, y) = \alpha \mathbf{X}_Y \tag{2}$$

where $\mathbf{X}'_Y$ is the adjusted pixel value, $\alpha$ is the scalar, and (x, y) indicates the pixel location in an $M \times N$ image. These operators can be formulated into the representation of an MLP:

$$\mathbf{Y} = \mathbf{R}(\mathbf{W}^T \mathbf{A} + b), \tag{3}$$

where $\mathbf{A} \in \mathbb{R}^{UV}$ is the vector flattened from the input. $\mathbf{W} \in \mathbb{R}^{UV \times UV}$ and $b \in \mathbb{R}^{UV}$ are weights and biases respectively, and $R(\cdot)$ is the activation function. Similarly, contrast adjustment refers to differences in luminance or color maps, the operation of which can also be expressed in MLPs [8]. To this end, we build an MLP-based Global Tone Perception module for adjusting the global brightness and contrast of an image and ensuring tonal consistency.

Specifically, given an input image $X$, we begin to obtain the lowest resolution LF component $\mathbf{L}_n$ of size $\frac{H}{2^n} \times \frac{W}{2^n}$ through multiple Gaussian down-sampling operations. Then, $\mathbf{L}_n$ is fed into the GTP module, which globally modulates the tones at the pixel level. The GTP is composed of three full convolutions, conditional network [8], and ReLU activation functions. The conditional network consists of three convolutions, ReLU activation, and an average pooling. The conditional network outputs a conditioning vector, which is used for global pixel-level modulation of the intermediate features of the base network, generating the lowest resolution global mapping image $\mathbf{L}_n^G$.

### 3.4 LOCAL TONE TUNING

Although global tone mapping yields visually consistent, it lacks modification of local details as well as contrast. Especially in areas with highlights or shadows, the single processing method of global mapping can result in some parts of the image appearing *flat* or *dead white*, which does not render light and shadow details well. For this reason, we introduce Local Tone Tuning, which enhances details in localized areas for better visual effects and details. The LTT includes a lightweight CNN network and several 3D LUTs. The CNN network is utilized to analyze the image context and predict

Table 1: Quantitative results of TM methods on the HDR+ dataset. The "N.A." result is unavailable due to insufficient GPU memory. The "*" symbol indicates that the results are adopted from the original paper (some are absent ("/")) due to the unavailable source code. Metrics with ↑ and ↓ denote higher better and lower better. The best and second results are in red and blue, respectively.

| Method | HDR+ (480p) | | | | | HDR+ (original 4K) | | | | |
|---|---|---|---|---|---|---|---|---|---|---|
| | PSNR↑ | SSIM↑ | TMQI↑ | LPIPS↓ | △E↓ | PSNR↑ | SSIM↑ | TMQI↑ | LPIPS↓ | △E↓ |
| UPE[25] | 23.33 | 0.852 | 0.856 | 0.150 | 7.68 | 21.54 | 0.723 | 0.821 | 0.361 | 9.88 |
| HDRNet[26] | 24.15 | 0.845 | 0.877 | 0.110 | 7.15 | 23.94 | 0.796 | 0.845 | 0.266 | 6.77 |
| CSRNet[8] | 23.72 | 0.864 | 0.884 | 0.104 | 6.67 | 22.54 | 0.766 | 0.850 | 0.284 | 7.55 |
| DeepLPF[3] | 25.73 | 0.902 | 0.877 | 0.073 | 6.05 | N.A. | N.A. | N.A. | N.A. | N.A. |
| LUT[5] | 23.29 | 0.855 | 0.882 | 0.117 | 7.16 | 21.78 | 0.772 | 0.850 | 0.303 | 9.45 |
| LPTN[23] | 24.80 | 0.884 | 0.885 | 0.087 | 8.38 | 24.05 | 0.807 | 0.839 | 0.207 | 9.04 |
| CLUT[13] | 26.05 | 0.892 | 0.886 | 0.088 | 5.57 | 24.04 | 0.789 | 0.848 | 0.245 | 6.78 |
| sLUT[20]* | 26.13 | 0.901 | / | 0.069 | 5.34 | 23.98 | 0.789 | / | 0.242 | 6.85 |
| SepLUT[7] | 22.71 | 0.833 | 0.879 | 0.093 | 8.62 | 21.87 | 0.731 | 0.842 | 0.220 | 9.52 |
| LLFLUT[4]* | 26.62 | 0.907 | / | 0.063 | 5.31 | 25.32 | 0.849 | / | 0.149 | 6.03 |
| CoTF[27] | 23.78 | 0.882 | 0.876 | 0.072 | 7.76 | 23.44 | 0.818 | 0.866 | 0.175 | 8.01 |
| Ours-Tiny | 27.59 | 0.925 | 0.886 | 0.035 | 5.87 | 27.13 | 0.902 | 0.867 | 0.065 | 5.88 |
| Ours | 27.85 | 0.927 | 0.887 | 0.036 | 5.68 | 27.90 | 0.917 | 0.869 | 0.061 | 5.32 |

the weights for multiple 3D LUTs. The CNN network consists of three convolutional layers with LeakyReLU and InstanceNorm, an average pooling layer, and a reshape operation. Multiple 3D LUTs are responsible for the local tone mapping of each patch for local detail refinement.

Specifically, we first interpolate $\mathbf{L}_n^G$ to the fixed-resolution version (e.g., $80 \times 60$) and convert the feature maps into a compact vector representation $\mathbf{E} \in N^2 \times d$ via CNN network. The $E$ represents the local attributes of the $N^2$ patches of the input image and $d$ indicates the number of sampling points. That is, we divide the image into $N^2$ patches. Then, we feed the $1 \times d$ local feature vectors of each patch into the 3D LUT prediction network, where the 3D LUT parameters are predicted by two fully connected layers. For the whole image, we obtain $N^2$ 3D LUT parameters for adjusting the tone of each patch. Following the completion of 3D LUT prediction for each region, due to the limited number of sampling points, we implement LUT mapping through trilinear interpolation. To address the edge discontinuity problem caused by chunking, we obtain the mapping values at the boundary by bilinear interpolation of the LUT mappings of the adjacent regions. Finally, we obtain the low-resolution mapped image $\mathbf{T}_n$ through the local tone tuning.

### 3.5 LEARNABLE DIFFERENTIAL PYRAMID

Compared to general images, HDR images are more difficult to extract high frequency due to a wider range of luminance, complexity of high-frequency details, and color accuracy and consistency. Existing methods struggle to recover these details and colors effectively, see in Fig. 8. We introduce a learnable differential pyramid module, inspired by scale-invariant feature transform [28, 29], for adaptively extracting high-frequency information from HDR images. The differential pyramid exhibits two key properties [29, 28] that make it suitable for tone mapping tasks. 1) The features within the differential pyramid remain stable under scale transformations, rotations, and illumination variations, enabling effective tone mapping across varying conditions. 2) The differential pyramid, constructed via filtering and difference operations, efficiently captures image edges, corners, and textures, which are crucial for preserving details.

In detail, for the input image $\mathbf{X}$, we first apply a $3 \times 3$ convolution to obtain the initial feature map $\mathbf{F}_0$. Subsequently, we generate a lower-resolution feature map $\mathbf{F}_1$ through three consecutive convolutions and a maximum pooling operation. We then compute the difference between three convolutions to obtain three sets of differential features, which are concatenated together and fed into a residual network to generate the first high-frequency feature $\mathbf{H}_0$. This process iterates, with differential feature maps generated through successive convolutions and pooling, which are concatenated and fed into the residual network to predict HF components. Through $n-1$ iterations, we obtain the complete differential pyramid $\mathbf{X}_{HF} = [H_0, H_1, \ldots, H_{n-1}]$ that contains multi-scale high-frequency features adaptively learned from the input HDR image, tapering resolutions from $H \times W$ to $\frac{H}{2^{n-1}} \times \frac{W}{2^{n-1}}$. $n$ denotes the number of pyramid levels ($n$=3 in our framework).

### 3.6 REFINEMENT ON HIGH-FREQUENCY COMPONENT

To achieve faithful reconstruction in tone mapping manipulation, this study proposes an iterative mask strategy to refine HF components $\mathbf{X}_{HF} = [H_0, H_1, ..., H_{n-1}]$. Specifically, we first up-sample the low-frequency mapped image $\mathbf{T}_n$ and concatenate it with HF component $\mathbf{H}_{n-1}$, then fed it into a

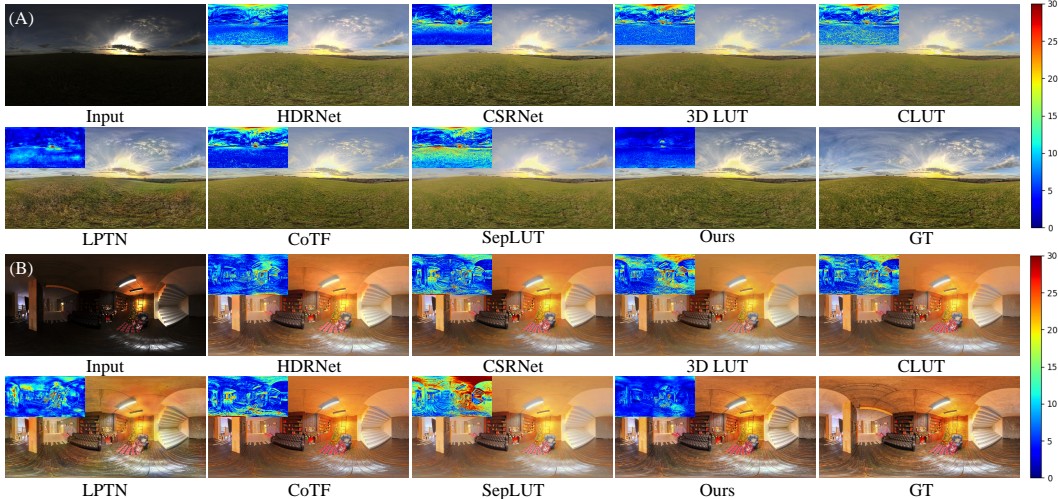

Figure 3: Visual comparisons between our DPRNet and the state-of-the-art methods on the HDRI Haven dataset (Zoom-in for best view). The error maps in the upper left corner facilitate a more precise determination of performance differences.

residual network to predict the mask $\mathbf{M}_{n-1}$. This mask allows pixel-by-pixel refinement of the HF component, which is subsequently added to the up-sampling $\mathbf{T}_n$ to generate the reconstructed result of the current layer $\mathbf{T}_{n-1}$. The operations at the $n_{th}$ layer can be formulated as:

$$\mathbf{M}_{n-1} = \text{Res}(\text{Up}(\mathbf{T}_n) \copyright \mathbf{H}_{n-1}) \tag{4}$$

$$\mathbf{T}_{n-1} = \text{Up}(T_n) + (\mathbf{H}_{n-1} \times \mathbf{M}_{n-1}) \tag{5}$$

where Res, Up$(\cdot)$, and $\copyright$ denote the residual network, up-sampling operation, and concatenation operation, respectively. This iterative process continues, and the mapped image $\mathbf{T}_{n-1}$ is fed to the next layer to further refine the HF component $\mathbf{H}_{n-1}$, and obtain LDR images $\mathbf{T}_{n-2}$. Through $n-1$ iterations, we progressively refine the high-frequency components, ultimately obtaining a high-quality LDR image $\mathbf{T}$. The residual network consists of a $3 \times 3$ convolution, a ReLU activation function, three residual blocks, and a cascade of $3 \times 3$ convolution.

## 3.7 LOSS FUNCTIONS

To maintain the accuracy of the reconstructed image, we jointly optimize the reconstruction with pixel-wise $L_{\text{Re}}$ and structural similarity $L_{\text{ssim}}$, high-frequency, and perceptual loss. To summarize, the complete objective of our proposed model is combined as follows:

$$L_{\text{total}} = \alpha \cdot L_{\text{Re}} + \beta \cdot L_{\text{ssim}} + \gamma \cdot L_{\text{HF}} + \eta \cdot L_{\text{p}} \tag{6}$$

$\alpha$, $\beta$, $\gamma$, and $\eta$ are the corresponding weight coefficients. More details are in the appendix.

## 4 EXPERIMENTS

### 4.1 EXPERIMENTAL SETTINGS

**Datasets.** We evaluate our method on four publicly available datasets: HDR+ Burst Photography [30], HDRI Haven [1], HDR Survey [31], and UVTM video dataset [32]. The HDR+ dataset is a staple for HDR reconstruction and tone mapping research, especially in mobile photography. Following Zeng et al.'s preprocessing method [5, 4], we utilize 675 image sets for training and 248 for testing at both 480p and 4k resolutions. The HDRI Haven dataset is widely recognized as one of the benchmarks for evaluating tone mapping [32, 33], includes 570 HDR images ($4096 \times 2048$) of diverse scenes under various lighting conditions. We select 456 image sets for training and 114 for testing. Similar to the HDR+ dataset preprocessing method, experiments are conducted at both 480p and 4k resolutions. The HDR Survey dataset consists of 105 HDR images, with no ground truth, and is one of the benchmarks for HDR tone mapping evaluations. [1, 16, 17, 34, 35]. The UVTM video dataset, also with no ground truth, includes 20 real captured HDR videos. Note that the HDR Survey and UVTM video datasets are only for testing purposes.

_______________________
[1] https://hdri-haven.com/

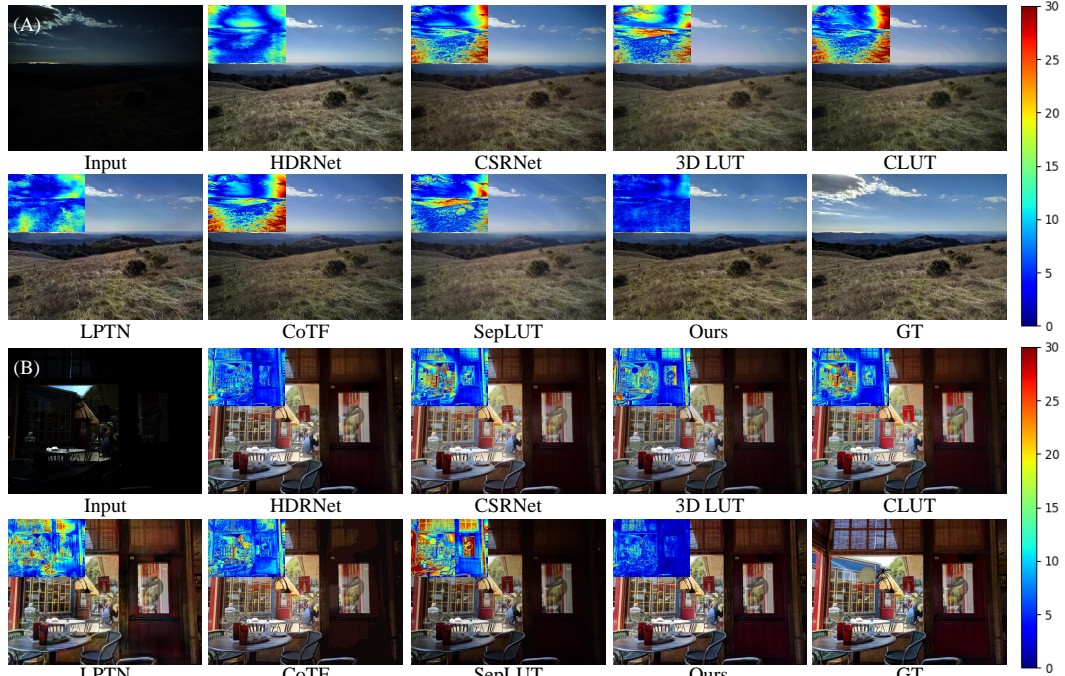

Figure 4: Visual comparisons between our DPRNet and the state-of-the-art methods on the HDR+ dataset (Zoom-in for best view). The error maps in the upper left corner facilitate a more precise determination of performance differences.

**Evaluation metrics.** We evaluate model performance in five dimensions: pixel accuracy, structural integrity, mapping quality, perceptual quality, and color fidelity. Specifically, we report traditional PSNR and SSIM metrics on the RGB channel to evaluate the reconstruction accuracy. We also employ TMQI [36] and LPIPS [37] to evaluate image mapping quality and perceptual quality respectively. For LPIPS, we use the AlexNet to extract feature maps. In addition, we use the CIELAB color space [38] (also known as $\Delta E$) to evaluate the difference in color dimensions of the reconstructed images.

**Implementation details.** We implement our model with Pytorch [39] on the RTX 3090 GPU platform. We have set the batch size to 1 (original 4K resolution) or 4 (480P). For optimization, we use AdamW optimizer [40] with $\beta_1 = 0.9$, $\beta_2 = 0.99$. The learning rate is initialized to $1 \times 10^{-4}$.

## 4.2 COMPARISON WITH STATE-OF-THE-ARTS

**Quantitative comparison.** We quantitatively compare the proposed method with a wide range of state-of-the-art tone mapping methods in Tab. 1, Tab. 2, and Tab. 7, in HDRNet [26], UPE [25], DeepLPF [3], CSRNet [8], LUT [5], CLUT [16], sLUT [20], LPTN [23], Sep-LUT [7], LLF-LUT [4], and CoTF [27]. Note that the input images considered in our evaluation are 16-bit uncompressed images in the CIE-XYZ color space, while the target images are 8-bit compressed images in the sRGB color space. Among these methods, DeepLPF, LPTN, and CSRNet are pixel-level methods based on ResNet and U-Net backbone, while HDRNet and UPE belong to patch-level methods, and 3DLUT, sLUT, Sep-LUT, LLF-LUT and CoTF are the image-level methods. Our method embraces the merits of these methods by designing joint global and local tone mapping. We follow the results in the published paper [4] and add more comparative methods, using publicly available source code and recommended configurations for training.

Our method significantly outperforms the SOTA methods LLF-LUT (NeurIPS 2023 [4]) and CoTF (CVPR 2024 [27]), improving PSNR by **2.58 dB** in the HDR+ and **3.31 dB** in the HDRI Haven dataset respectively. Tab. 1 shows the quantitative comparison results on the HDR+ dataset at two different resolutions. Notably, our method exhibits a significant performance advantage over all competing methods on both resolutions, highlighted in red bold across all metrics. Benefiting from the learnable differential pyramid and iterative mask strategy, this advantage becomes more pronounced at the more challenging 4K resolution. Compared with the LLF-LUT, our DPRNet achieves notable improvements by 2.58 dB, 0.068, 0.088, and 0.71 in PSNR, SSIM, LPIPS, and $\Delta E$, respectively, which demonstrates the robustness of our method to high-resolution images. Moreover,

Table 2: Quantitative results of tone mapping methods on the HDRI Haven dataset. The "N.A." result is not available due to insufficient GPU memory.

| Method | HDRI Haven (480p) | | | | | HDRI Haven (original 4K) | | | | |
|---|---|---|---|---|---|---|---|---|---|---|
| | PSNR↑ | SSIM↑ | TMQI↑ | LPIPS↓ | △E↓ | PSNR↑ | SSIM↑ | TMQI↑ | LPIPS↓ | △E↓ |
| UPE[25] | 23.58 | 0.821 | 0.917 | 0.191 | 10.85 | 21.62 | 0.776 | 0.875 | 0.232 | 13.17 |
| HDRNet[26] | 25.33 | 0.912 | 0.941 | 0.113 | 7.03 | 23.61 | 0.899 | 0.890 | 0.143 | 8.19 |
| DeepLPF[3] | 24.86 | 0.939 | 0.948 | 0.077 | 7.64 | N.A. | N.A. | N.A. | N.A. | N.A. |
| CSRNet[8] | 25.78 | 0.872 | 0.928 | 0.153 | 6.09 | 24.42 | 0.863 | 0.875 | 0.174 | 6.83 |
| LUT[5] | 24.52 | 0.846 | 0.912 | 0.171 | 7.33 | 21.80 | 0.823 | 0.849 | 0.197 | 8.49 |
| CLUT[13] | 24.29 | 0.836 | 0.908 | 0.169 | 7.08 | 22.32 | 0.765 | 0.842 | 0.281 | 9.31 |
| LPTN[23] | 26.21 | 0.941 | 0.954 | 0.113 | 8.82 | 23.83 | 0.899 | 0.932 | 0.158 | 10.09 |
| SepLUT[7] | 24.12 | 0.854 | 0.915 | 0.165 | 8.03 | 22.79 | 0.838 | 0.859 | 0.180 | 9.11 |
| CoTF [27] | 26.65 | 0.935 | 0.948 | 0.098 | 5.84 | 24.58 | 0.891 | 0.911 | 0.156 | 7.19 |
| Ours-Tiny | 29.48 | 0.961 | 0.957 | 0.028 | 5.23 | 27.57 | 0.935 | 0.937 | 0.042 | 5.98 |
| Ours | **29.80** | **0.961** | **0.957** | **0.029** | **4.88** | **27.89** | **0.936** | **0.939** | **0.042** | **5.46** |

our proposed tiny version also outperforms the SOTA method by 1.81 dB on the HDR+ dataset. Similarly, evaluation on the HDRI Haven dataset presents an even more pronounced advantage, with an improvement of more than 3.3 dB at 4K resolution, especially in the LPIPS metric, which is far superior to all existing methods, thanks to our finely designed tone mapping framework.

**Qualitative results.** Visual comparison of DPRNet and state-of-the-art tone mapping methods are shown in Fig. 3, Fig. 4, and Fig. 6. Please zoom in for better visualization. To better visualize the performance differences of various methods, we present an error map to show the differences between the results of each method and the target image, as shown in the upper left corner of the image. In the error map, the red area indicates a larger difference, while the blue area indicates that the two are closer. These figures illustrate that our DPRNet consistently delivers visually appealing results on the HDR+ and HDRI Haven datasets. For instance, in Fig. 4, our method

Table 3: Compare computational efficiency.

| Methods | #Params | MACs | Runtime |
|---|---|---|---|
| UPE | 999K | 1.2G | 133.2ms |
| HDRNet | 482K | 1.1G | 4.56ms |
| LUT | 592K | 0.7G | 2.7ms |
| CSRNet | 37K | 52.8G | 48.3ms |
| sLUT | 4.52M | 113.8G | / |
| CLUT | 952K | 9.4G | 8.4ms |
| DeepLPF | 1.72M | 454.4G | N.A. |
| UnpairedTM | 4.45M | 1410G | 162.3ms |
| Ours | 818K | 155G | 104.6ms |

exhibits excellent local detail processing capability while maintaining excellent color fidelity with a better balance between global and local areas. In contrast, previous methods either result in color distortion (e.g., 3D LUT, SepLUT, and LPTN in Fig. 4 (A), LPTN, CoTF in Fig. 3 (A)) or fail to map local area (e.g., HDRNet, CSRNet, SepLUT, CoTF in Fig. 4 (B), CLUT, 3D LUT in Fig. 3 (B)). Additionally, our method faithfully reconstructs fine high-frequency textures while ensuring precise color reconstruction and vivid color saturation. More visual comparisons are presented in our supplementary material and https://xxxxxx2024.github.io/DPRNet/.

**Computational efficiency.** To demonstrate the practicality of the proposed method, we evaluate the inference time on 114 images and the number of parameters, the multiply-accumulate operations (MACs), and the runtime. All presented results are obtained using a 32GB NVIDIA V100 GPU. Multiply-accumulate operations (MACs) are computed based on the input dimensions $3840 \times 2160 \times 3$. Tab. 3 shows that our DPRNet strikes a good balance between efficiency and performance.

### 4.3 Ablation studies

**Effectiveness of specific modules.** To validate the effectiveness of the GTP and LTT modules in the LF component, we remove the two modules individually. We remove both GTP and LTT modules from the proposed DPRNet as the baseline network (w/o TM). Tab. 4 presents the results of different variants. When both GTP and LTT are removed, the model performance shows severe degradation, by 3.33 dB PSNR. Meanwhile, the PSNR decreases by more than 1 dB regardless of either removing GTP or LTT alone, which indicates that a single GTP or LTT is incapable of reconstructing satisfactory tone mapping results. This corroborates the efficacy of our joint GTP and LTT strategy. As illustrated in Fig. 5, the integration of GTP with LTT yields promising visual quality in both global and local enhancement in this challenging case. These results provide compelling evidence that our proposed joint GTP and LTT framework outperforms existing TM methods.

To verify the effectiveness of the learnable differential pyramid, we replace our LDP module with single down-sampling, the Gaussian pyramid (GP), and Laplacian pyramid (LP), respectively. Tab. 4 shows the results for different variants. When LDP is introduced, the results are improved by 3.1 dB.

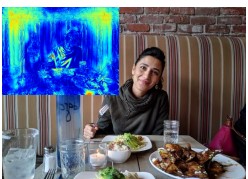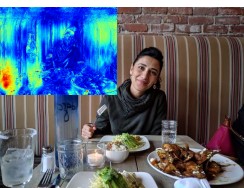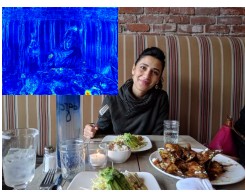

| W/O TM | W/O LTT | W/O GTP | Original Result |

Figure 5: Visual results of ablation study on framework component.

Table 4: Ablation study on the global and local tone mapping modules.

| Metrics | LF Tone Mapping | | | Pyramid | | | Refine HF | Original Result |
|---|---|---|---|---|---|---|---|---|
| | w/o TM | w/o LTT | w/o GTP | w/o HF | GP | LP | w/o Ref. | Replacement |
| PSNR | 24.52 | 26.52 | 26.20 | 21.74 | 22.77 | 24.71 | 26.47 | **27.85** |
| SSIM | 0.915 | 0.914 | 0.877 | 0.579 | 0.838 | 0.843 | 0.873 | **0.927** |
| TMQI | 0.869 | 0.877 | 0.878 | 0.854 | 0.874 | 0.879 | 0.870 | **0.887** |
| LPIPS | 0.039 | 0.039 | 0.038 | 0.286 | 0.122 | 0.117 | 0.072 | **0.036** |
| △E | 7.18 | 6.47 | 0.039 | 9.45 | 12.32 | 10.34 | 6.78 | **5.68** |

This evidence highlights the success of the learnable differential pyramid module. Meanwhile, the results show that GP or LP is not able to effectively extract the high-frequency information of HDR images, resulting in the presence of significant degradation. When we use an iterative mask strategy to refine the high-frequency (compared to w/o Ref.), the result is further improved by 1.3 dB. This finding demonstrates that the iterative mask strategy helps to produce a finer high-frequency texture.

**Selection of the number of levels.** We validate the influence of the number of pyramid levels $n$. As shown in Tab. 5, the model achieves the best performance on all tested resolutions when $n = 4$. When a larger number of levels ($n \geq 5$) result in a significant decline in performance. This is because when $n$ is larger and the number of downsamples is more, the model fails to reconstruct the high frequencies efficiently, resulting in performance degradation.

Table 5: Ablation study on the pyramid levels numbers and the number of grids in the LTT.

| Metrics | Number of Levels | | | Grid Size | | | | | |
|---|---|---|---|---|---|---|---|---|---|
| | n=3 | n=4 | n=5 | N=2 | N=4 | N=6 | N=8 | N=12 | N=16 |
| PSNR | 27.77 | **27.85** | 26.33 | 26.08 | 27.45 | 27.67 | **27.85** | 27.24 | 27.38 |
| SSIM | 0.925 | **0.927** | 0.916 | 0.919 | 0.918 | 0.925 | **0.927** | 0.923 | 0.924 |
| TMQI | 0.880 | **0.887** | 0.877 | 0.878 | 0.880 | 0.878 | **0.887** | 0.876 | 0.878 |
| LPIPS | 0.036 | **0.036** | 0.040 | 0.040 | 0.037 | 0.036 | **0.036** | 0.036 | 0.036 |
| △E | 5.80 | **5.68** | 6.29 | 6.58 | 5.92 | 5.81 | **5.68** | 6.03 | 5.98 |
| #Params | **2.92M** | 2.93M | 2.97M | **287K** | 818K | 1.70M | 2.93M | 6.48M | 11.4M |

**Selection of the number of grids.** Furthermore, we verify the effect of grid number on tone mapping in local tone tuning. As shown in Tab. 5, the model achieves optimal performance when $N = 8$. When we reduce the number of grids (i.e., when $N = 4$ is Ours-Tiny version), the number of model parameters also decreases. The proposed framework still significantly outperforms the existing SOTA method, the parameters are reduced from 2.9M to 818K and the PSNR is slightly reduced by 0.4 dB. There is no continuous increase in performance when N is greater than 8 when local tone mapping gradually tends to global mapping.

**Ablation study on loss functions.** To test the effect of the loss function, we set up different variants and modified the loss function combination step by step. Tab. 6 shows Variants #1 and #2 improve performance about 0.2dB. In particular, Variants #3 shows that the addition of $L_p$ loss results in 0.86 PSNR higher than the baseline. Variants #4 show a 0.13 dB performance improvement by adding high-frequency loss $L_{HF}$.

Table 6: Ablation study on the loss function.

| Variants | $L_{Re}$ | $L_{ssim}$ | $L_{HF}$ | $L_p$ | PSNR↑ | SSIM↑ |
|---|---|---|---|---|---|---|
| #1 | ✓ | ✗ | ✓ | ✓ | 27.60 | 0.922 |
| #2 | ✓ | ✓ | ✗ | ✓ | 27.72 | 0.925 |
| #3 | ✓ | ✓ | ✓ | ✗ | 26.99 | 0.913 |
| #4 | ✓ | ✓ | ✓ | ✓ | 27.85 | 0.927 |

### 4.4 USER STUDY

To evaluate the overall performance of the photorealism and visual quality, we perform a user study based on human perception. We have conducted a user study with 25 participants for subjective evaluation. The participants are asked to rank four tone mapped images (HDRNet [26], CLUT [13],

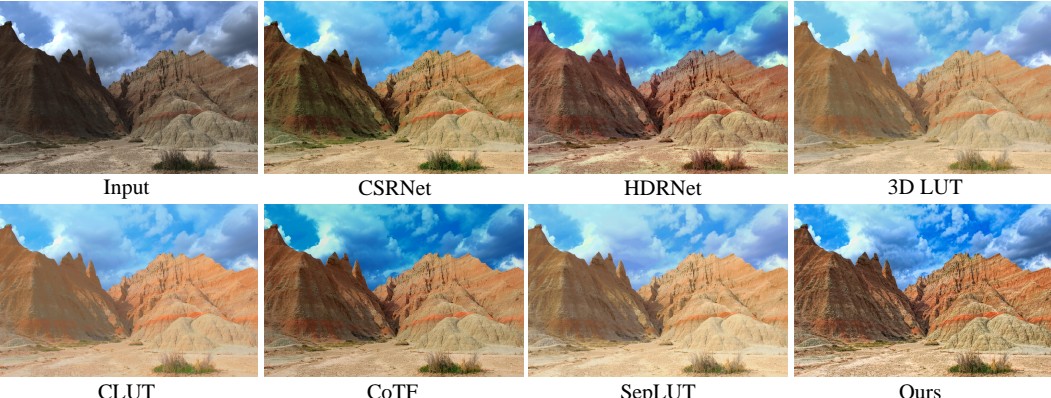

| Input | CSRNet | HDRNet | 3D LUT |
| CLUT | CoTF | SepLUT | Ours |

Figure 6: Visual comparisons between our DPRNet and the state-of-the-art methods, training on the HDRI Haven dataset, and testing on the HDR Survey dataset.

GT, and ours) according to the aesthetic visual quality. 50 images are randomly selected from the testing set and are shown to each participant. Four tone mapping results are displayed on the screen in a random order. Users are asked to pay attention to the color, details, artifacts, and whether the local color is harmonious. As shown in Fig. 7, our results achieve a better visual ranking compared to HDRNet and CLUT. Also, we got 423 images ranked first and 562 images ranked second, accounting for 39.4% of the total, achieving results comparable to the ground truth (accounting for 39.4%).

## 4.5 VALIDATION OF GENERALIZATION

We evaluate the generalization performance of the proposed model on the HDR Survey dataset [31] and UVTM video dataset [32] using the no-reference metric TMQI as the evaluation metric. We use the models trained on the HDR Haven dataset and report the results in Tab. 7. Our proposed model achieves the highest TMQI scores and significantly outperforms the other methods. Compared with the CoTF, DPRNet achieves notable improvements of 0.0389, demonstrating its superiority in handling HDR content across diverse scenarios and datasets. Additionally, the results validate the generalization performance and practical application of our DPRNet. Visual results in Fig. 6 show the superiority of the generalization performance of our model.

Table 7: Validating generalization on third-party datasets include HDR Survey and UVTM video datasets.

| Methods | TMQI | |
| | HDR Survey | UVTM |
| --- | --- | --- |
| HDRNet [26] | 0.8641 | 0.8281 |
| CSRNet [8] | 0.8439 | 0.8973 |
| 3D LUT [5] | 0.8165 | 0.8787 |
| CLUT [13] | 0.8140 | 0.8799 |
| LPTN [23] | 0.8964 | 0.9141 |
| SepLUT [7] | 0.8085 | 0.8629 |
| IVTMNet [32] | 0.9160 | 0.8991 |
| CoTF [27] | 0.8612 | 0.9006 |
| Ours-Tiny | 0.9223 | **0.9395** |
| Ours | **0.9244** | 0.9364 |

Figure 7: Ranking results of user study. Rank 1 means the best visual quality.

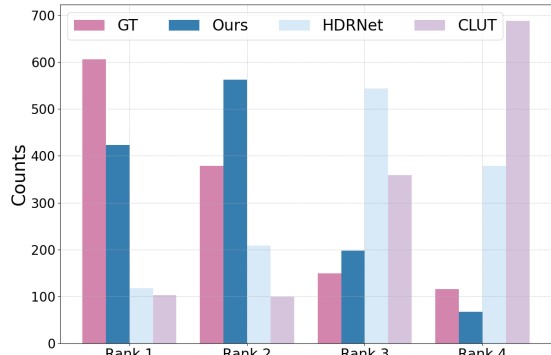

## 5 CONCLUSION

This paper proposes a learnable Differential Pyramid Representation Network (DPRNet), a comprehensive framework capable of recovering details and manipulating global and local tone mappings. We construct a Global Tone Perception module to modulate the image globally. Then the Local Tone Tuning generates TMO coefficients at the patch level to refine the local tones. Furthermore, we propose a Learnable Differential Pyramid module to capture multi-scale high-frequency components, coupled with an Iterative Detail Enhancement progressively refining image resolution and image details. Extensive experiments demonstrate that our method significantly outperforms state-of-the-art methods, improving PSNR by 2.58 dB in the HDR+ dataset and 3.31 dB in the HDRI Haven dataset respectively compared with the second-best method. In addition, our method has the best generalization ability in unsupervised video TMO.

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

# Appendix

In this Appendix, we present the related work and provide additional results and analysis.

## A   LOSS FUNCTIONS

The proposed framework obtains faithful global and local enhancement by optimizing the reconstruction loss, perceptual loss, and high-frequency loss.

### A.1   RECONSTRUCTION LOSS

To maintain the accuracy of the reconstructed image, we directly adopt pixel-wise $L_{Re}$ and $\mathcal{L}_{\text{ssim}}$ loss on the prediction $T$ and the ground truth $Y$:

$$\mathcal{L}_{Re} = \sum_{i=1}^{n} \left\| T_i - Y_i^{LF} \right\|_1 + \left\| T - Y \right\|_1 , \tag{7}$$

$$\mathcal{L}_{ssim} = 1 - \text{MS-SSIM}(T, Y). \tag{8}$$

where $T_i$ denotes the output of each layer of the network and $Y_i$ denotes the Gaussian pyramid of the ground truth.

### A.2   HIGH-FREQUENCY LOSS

To prompt the learnable differential pyramid module to extract effective high-frequency information, we introduce a high-frequency loss function. By calculating the $L_1$ loss between the output high-frequency component and the high-frequency of ground truth:

$$\mathcal{L}_{HF} = \sum_{i=0}^{n-1} \left\| H_i - Y_i^{HF} \right\|_1 , \tag{9}$$

where $Y_i^{HF}$ denotes the high-frequency component of the ground truth obtained through the Laplacian pyramid.

### A.3   PERCEPTUAL LOSS

To make the learned DPRNet more stable and robust, we employ a perceptual loss function that assesses a solution concerning perceptually relevant characteristics (e.g., the structural contents and detailed textures):

$$L_p = |\varphi^i(T) - \varphi^i(Y)\|_2^2, \tag{10}$$

where $\varphi$ represents the 5th convolution (before maxpooling) layer within VGG19 network [41].

### A.4   OUTPUT LOSS

To summarize, the complete objective of our proposed model is combined as follows:

$$L_{total} = \alpha \cdot L_{Re} + \beta \cdot L_{ssim} + \gamma \cdot L_{HF} + \eta \cdot L_p \tag{11}$$

$\alpha$, $\beta$, $\gamma$, and $\eta$ are the corresponding weight coefficients.

## B   COMPARISON OF LAPLACIAN PYRAMID AND LEARNABLE DIFFERENTIAL PYRAMID

To verify the efficacy of our proposed learnable differential pyramid (LDP), we have conducted a visual comparison of the high-frequency components extracted by LDP with those extracted by the Laplacian pyramid. As shown in Fig. 8, we can observe that when the dynamic range of the scene is sufficiently large, the Laplacian pyramid (LP) is unable to extract the image's high-frequency information efficiently. Conversely, our LDP exhibits enhanced robustness in this regard.

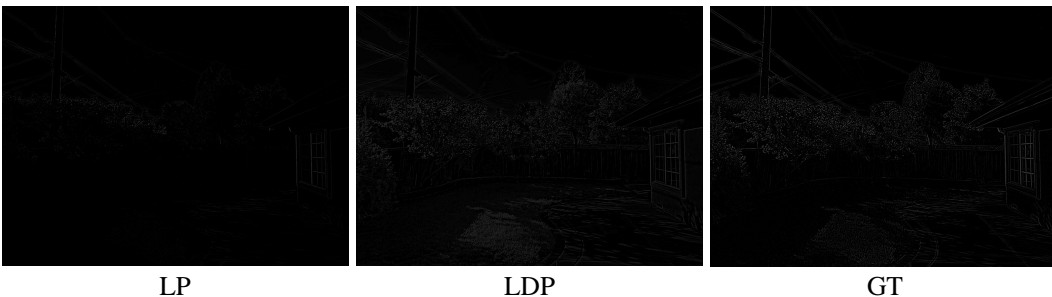

|          LP          |          LDP          |          GT          |

Figure 8: Comparison of differences in high-frequency components of input HDR images extracted by different modules.

## C    DIFFERENCES IN GENERALIZATION BETWEEN HDR+ DATASET AND HDRI HAVEN DATASET

To verify the difference in generalization between the two datasets, we use the HDR+ data and the HDRI Haven dataset to train separately and test on the HDR Survey dataset [42]. We use the no-reference metric TMQI as the evaluation metric. Tab. 8 shows that training on the HDRI Haven dataset is more suitable for the tone mapping task than HDR+, providing a new benchmark for subsequent tone mapping tasks. Meanwhile, the results in tab. 8 shows that our proposed DPRNet has stronger model generalization performance.

Table 8: Differences in generalization between HDR+ Dataset and HDRI Haven dataset.

| Train Dataset | Test Dataset | TMQI | | | | | |
|---|---|---|---|---|---|---|---|
| | | CSRNet | 3D LUT | CLUT | LPTN | CoTF | Ours |
| HDR+ [30] | HDR Survey | 0.7754 | 0.7847 | 0.7638 | 0.7892 | 0.7959 | 0.8142 |
| HDRI Haven | HDR Survey | 0.8439 | 0.8165 | 0.8140 | 0.8964 | 0.8612 | 0.9244 |

**Ablation study on size of the 3D LUTs.** For verification, we ablate the width of the backbone network by varying the hyper-parameter m under the same setting of the LUT sizes (Soand St) and report the results in Tab. 9. The ablation results indicate that increasing the width of the backbone does not guarantee enhanced performance but might increase the capacity redundancy and the training difficulty.

Table 9: Ablation study on size of the 3D LUTs.

| M | S | PSNR | SSIM | #Params | Runtime |
|---|---|---|---|---|---|
| 6 | 9 | 27.70 | 0.928 | 212K | 8.57ms |
| 8 | 9 | 27.74 | 0.918 | 214K | 8.69ms |
| 6 | 17 | 27.79 | 0.925 | 815k | 8.94ms |
| 8 | 17 | 27.85 | 0.927 | 818k | 9.16ms |

## D    MORE QUALITATIVE RESULTS

In this section, we provide additional qualitative comparisons on UVTM video, HDRI Haven, HDR+, and HDR Survey datasets. Fig. 9 to Fig. 14 show 14 sets of qualitative comparisons. It is observed that DPRNet successfully produces outputs with finer details and sharper edges. Furthermore, with the proposed joint global and local tone mapping, DPRNet further improves the color saturation of the output results. It can be observed that our proposed method achieves the best TMQI scores. For TMQI, our method outperforms the second-best method [23] by 0.0254 on the UVTM dataset. It demonstrates that our method is good at revealing details and keeping temporal consistency. Fig. 14 shows four sets of qualitative comparisons on the HDR Survey dataset. From the examples, we see that DPRNet can restore the fine details, leading to plausible results. Notably, we train on the HDRI

# E   DPRNet for HDR video tone mapping

In this section, we further demonstrate the effectiveness and model generalization performance of the proposed DPRNet for video tone mapping. We use the HDRI Haven image dataset for training and testing on the UVTM video dataset [43]. For display purposes, we have created an anonymous web page showing our results on the video dataset. As can be observed from the four sets of videos shown, our method generates videos with suitable contrast between light and dark and does not suffer from any unnatural textures. Our method exhibits perfect tone mapping results in different sophisticated scenes, proving the effectiveness of our proposed DPRNet and the strong model generalization ability.

# F   Limitations and discussion

Our method does have limitations. For the MIT FiveK dataset, our method is unable to obtain SOTA results, and we have performed some analyses on this. Firstly, FiveK was shot by a DSLR camera, which was not processed by advanced ISP pipeline and lacked raw denoising and YUV denoising, resulting in the dataset containing serious noise. These noises have a great influence on our high-frequency extraction. Second, some reference images in the FiveK dataset suffer from overexposure or oversaturation, which poses a challenge to the enhancement method, as mentioned by Zhang et al [4]. Thirdly, there are inconsistencies in the reference images adjusted by the same professional photographer, leading to differences between the training and test sets.

As a future work, we plan to build a unified model to tackle more visual enhancement tasks by integrating denoising, and tone mapping.

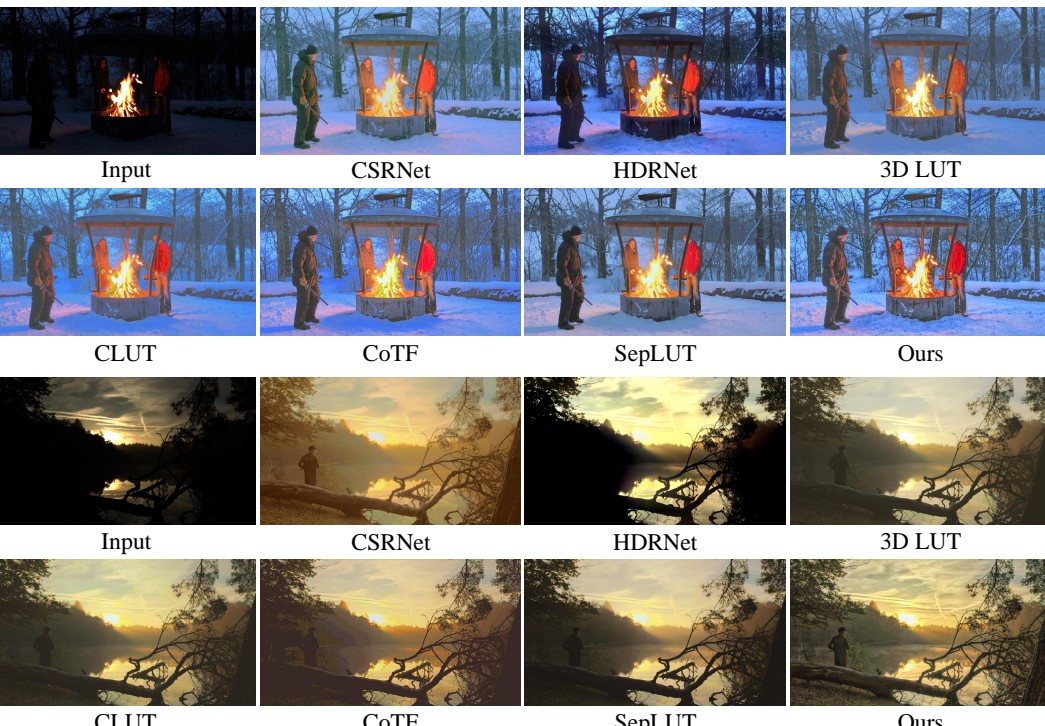

Figure 9: Visual comparisons between our DPRNet and the state-of-the-art methods, training on the HDR Haven dataset, and testing on the UVTM video dataset [32].

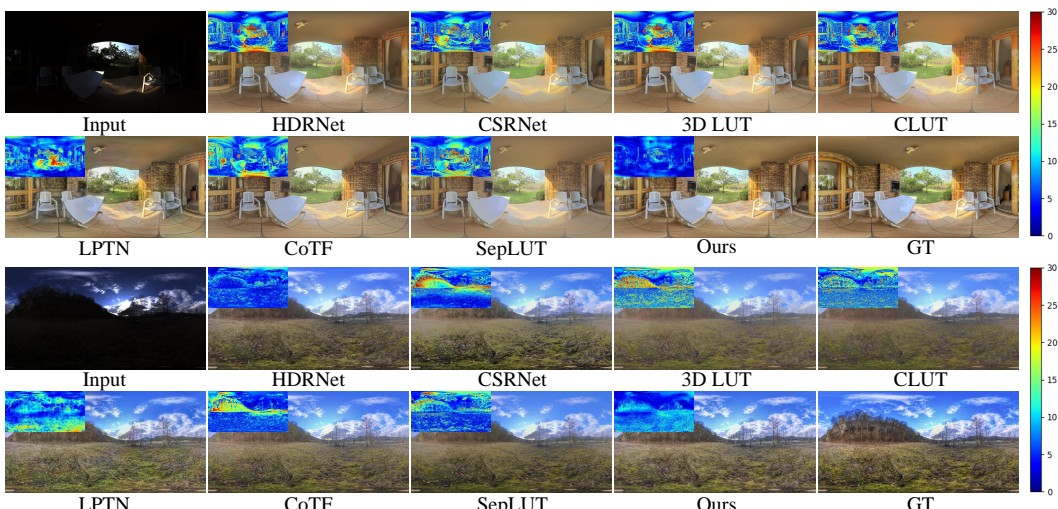

Figure 10: Visual comparisons between our DPRNet and the state-of-the-art methods on the HDRI Haven dataset (480p resolution).

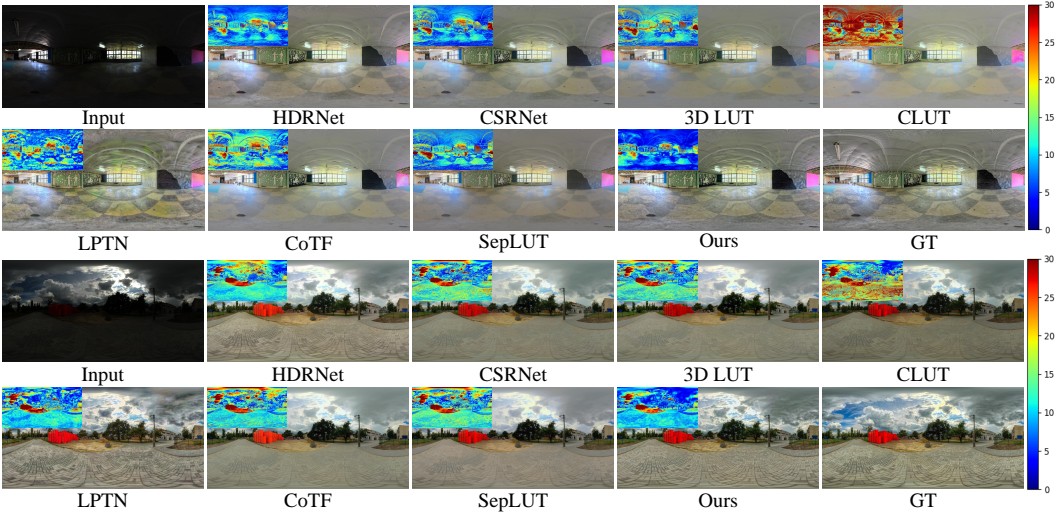

Figure 11: Visual comparisons between our DPRNet and the state-of-the-art methods on the HDRI Haven dataset (4K resolution).

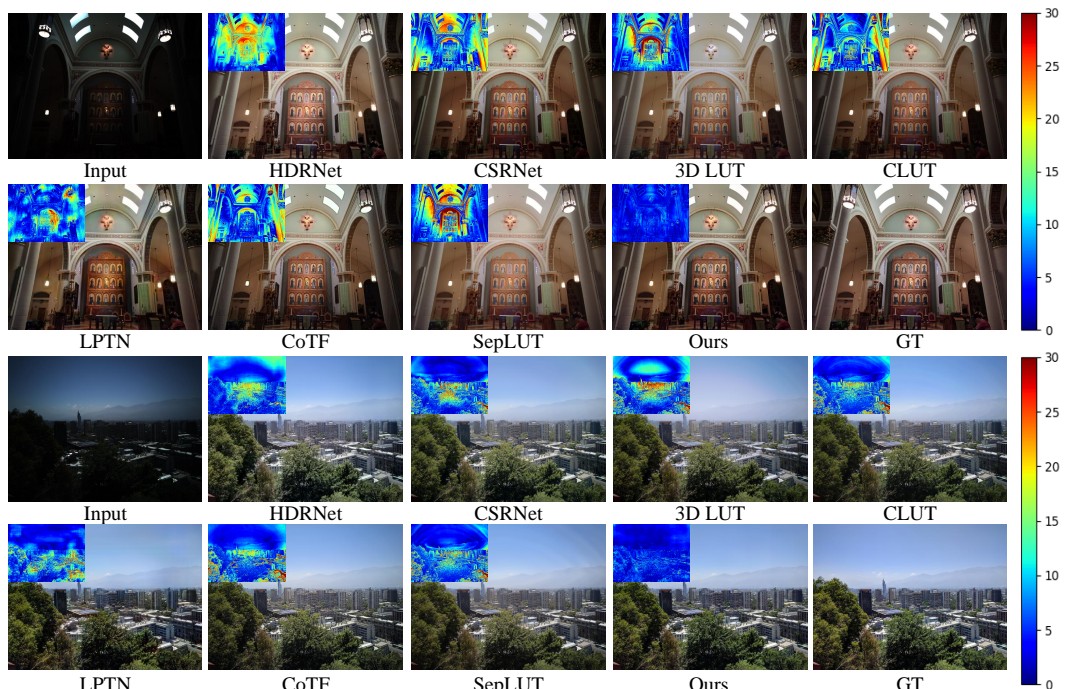

Figure 12: Visual comparisons between our DPRNet and the state-of-the-art methods on the HDR+ dataset (480p resolution).

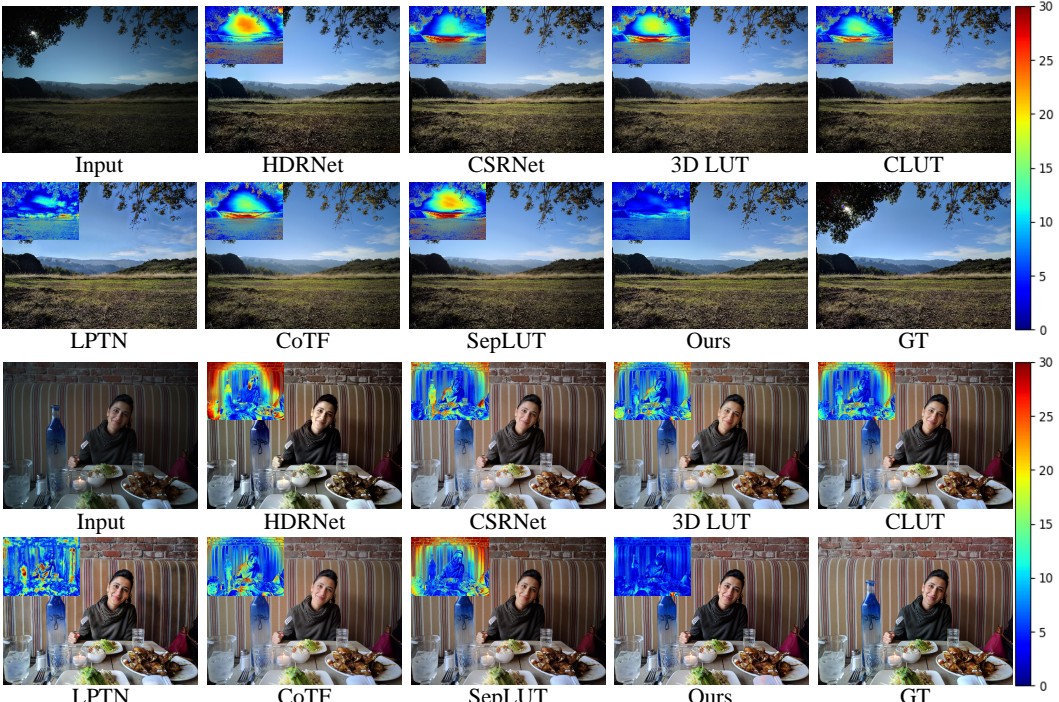

Figure 13: Visual comparisons between our DPRNet and the state-of-the-art methods on the HDR+ dataset (4K resolution).

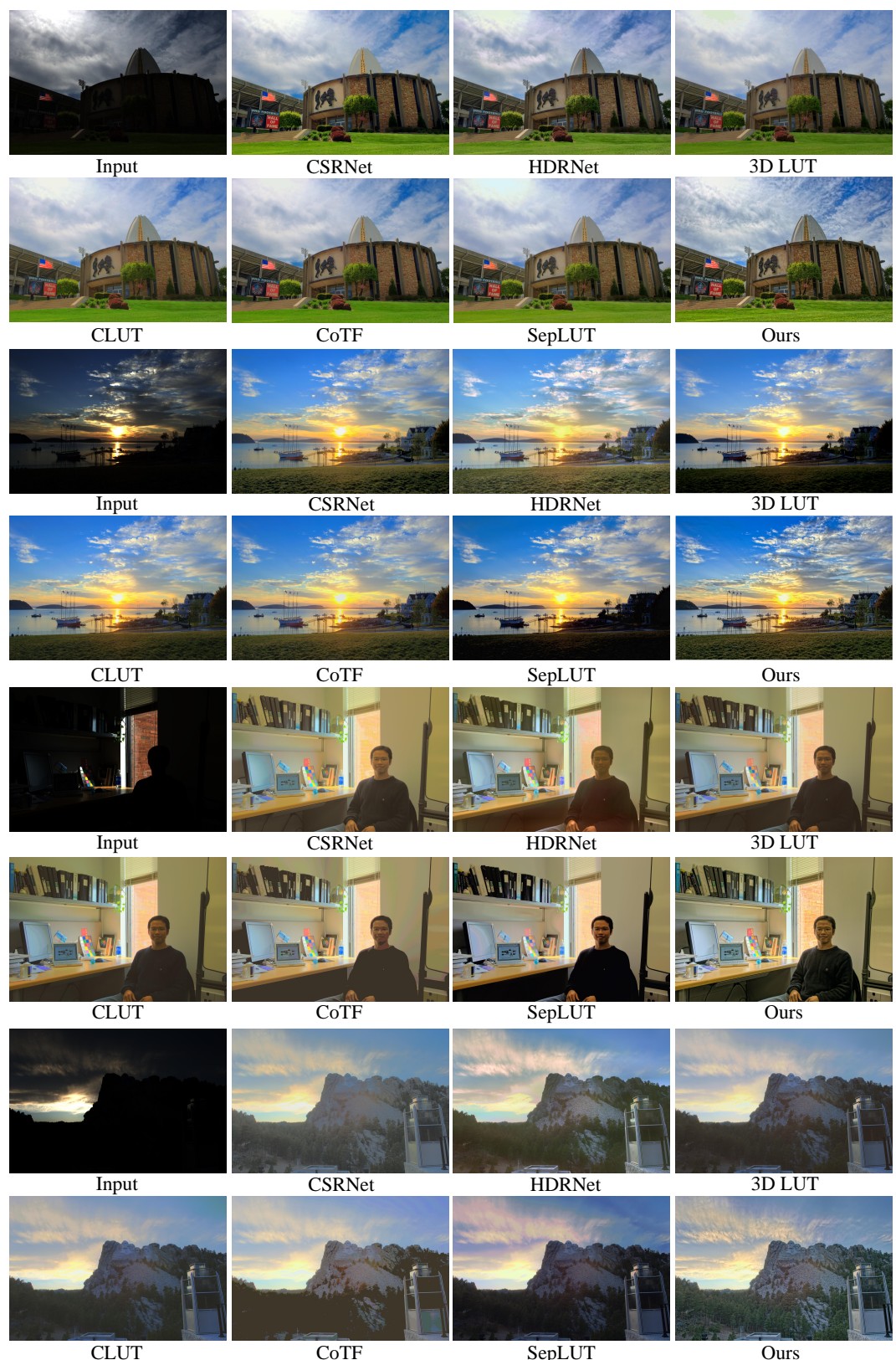

Figure 14: Visual comparisons between our DPRNet and the state-of-the-art methods, training on the HDR Haven dataset, and testing on the HDR Survey dataset.

