# OpenReview forum: "Learning Differential Pyramid Representation for Tone Mapping"
_ICLR.cc/2025/Conference — ICLR 2025 Conference Withdrawn Submission_

### Official Review · Reviewer_VAnv · 2024-10-28

**Soundness:** 2
**Presentation:** 2
**Contribution:** 2
**Rating:** 3
**Confidence:** 4

**Summary:**

This paper proposes an end-to-end network named Differential Pyramid Representation Network (DPRNet), in which Global Tone Perception (GTP) module and Local Tone Tuning (LTT) module conduct global and local adjustments, and Learnable Differential Pyramid (LDP) and Iterative Detail Enhancement (lDE) module extract high-frequency features and restore details. Extensive experiments show that this method outperforms previous methods.

**Strengths:**

1. Some modules have interesting designs. The Local Tone Tuning (LTT) module learns different LUTs for different areas to accomplish local adjustments, and avoids edge discontinuity through bilinear interpolation.

2. Quantitative and qualitative experiments on HDR+ dataset achieve leading results. User study further enhances the credibility of the experiments.

**Weaknesses:**

1. There are textual overlaps with [1] and [2]. Lines 186-199 in this submission are identical to Section 3.1 in [1] except for notations. Line 287-288 is identical to the caption of Figure 4 in [2]. Lines 363-366 are nearly identical to Section 3.2 in [2]. All of them are without proper citation.

2. The Learnable Differential Pyramid (LDP) module and Iterative Detail Enhancement (IDE) module raise concerns, and please refer to the Questions Section.

3. The numeric citations do not follow the formatting requirements.

4. Some minor errors: The circled $C_c$ notation in Figure 2 differs from the circled $C$ in Equal 4. The lines in Table 4 are misaligned. Reconfirm the usage of uppercase of "W/O" in Figure 5.

[1] He, Jingwen, et al. "Conditional sequential modulation for efficient global image retouching." *Computer Vision–ECCV 2020: 16th European Conference, Glasgow, UK, August 23–28, 2020, Proceedings, Part XIII 16*. Springer International Publishing, 2020.

[2] Zhang, Feng, et al. "Lookup table meets local laplacian filter: pyramid reconstruction network for tone mapping." *Advances in Neural Information Processing Systems* 36 (2024).

**Questions:**

1. The high-frequency feature $H_n$ in LDP needs a clear definition. Section B of the Appendix indicates that $H_n$ is extracted from the HDR image and has a corresponding GT. Now that the GT of $H_n$ can be extracted from the input HDR image, why do we still need to use a network to learn $H_n$? It makes the intention of LDP confusing.

2. The inputs of IDE module are $H_n$ and $T_{n+1}$, and the output is $T_n$. But in Figure 2, the input of the first IDE module is only $H_0$, and the same goes for the second IDE module. Are the connection paths of $T$ omitted in Figure 2?

---

### Official Review · Reviewer_rsNJ · 2024-10-31

**Soundness:** 3
**Presentation:** 3
**Contribution:** 2
**Rating:** 5
**Confidence:** 4

**Summary:**

The authors focus on the HDR-to-LDR problem and propose a learnable Differential Pyramid Representation Network (DPRNet). The paper construct a Global Tone Perception module to modulate the image globally. Then the Local Tone Tuning generates TMO coefficients at the patch level to refine the local tones. To capture multi-scale high-frequency components, they propose a Learnable Differential Pyramid module to capture multi-scale high-frequency components, coupled with an Iterative Detail Enhancement progressively refining image resolution and image details. Extensive experiments demonstrate that the method significantly outperforms state-of-the-art methods.

**Strengths:**

1. The paper has a clear structure and high writing quality.
2. The experiments in the paper are very sufficient, and a large number of experiments are done to prove the effectiveness of the proposed module.
3.The experimental results of the proposed are excellent and surpass other methods.

**Weaknesses:**

1. Pyramid decomposition has been applied in [23][4]. For the low-resolution enhancement part, the method proposed in this paper is like a combination of CSRNet and 3DLUT, which seems to have limited innovation. In addition, this paper does not give full play to the speed advantage of pyramid decomposition.
2. The processing speed of the proposed method is slow, which is nearly 40 times slower than the LUT method.

**Questions:**

1. In the ablation experiment, can the effect gain brought by different modules offset the speed reduction?
2. The paper mainly compares some image enhancement algorithms. Is there a special HDR-to-LDR algorithm for comparison? And why is there no comparison [4] ?
3. This method is based on supervision. I don’t quite understand the meaning of “generalization ability in unsupervised video TMO” in the conclusion.
4. The tone mapping in the title of the paper is quite broad. It is recommended to focus on the problem solved in this paper.

---

### Official Review · Reviewer_rVr5 · 2024-11-01

**Soundness:** 2
**Presentation:** 2
**Contribution:** 2
**Rating:** 5
**Confidence:** 5

**Summary:**

The paper introduces a method to compress the high dynamic range content on low dynamic range display named learnable differential pyramid representation network. The network consists of a global tone perception, a local tone tuning, and learnable differential pyramid modules.

**Strengths:**

The paper is well written, proposed method network seems to work better than previous methods.

**Weaknesses:**

1. The reviewer can see that the paper lacks of literature review. For example, for tone mapping

- Unsupervised HDR Image and Video Tone Mapping via Contrastive Learning
- Fast global tone mapping for high dynamic range compression
- A Real-Time Semi-Supervised Deep Tone Mapping Network
- A Low-Latency Noise-Aware Tone Mapping Operator for Hardware Implementation with a Locally Weighted Guided Filter

2. Model
- The design and the explanation of the model is confusing. The reviewer find that the explanation does not match the figure.
- It is not clear which one is the main contribution in the model, the reviewer believe that the LDP is the main, others are employed from other papers.

3. The paper lacks of a strong motivation of the paper. The reviewer understand that the target is the LDR display, but did not see that when reading the paper.

4. The reviewer understand that the paper follow [4] when comparing with other methods. However, the reviewer think that comparing a ToneMapping method with image enhancement methods is not fair. If the author did that, were they retrained with the dataset that used in the proposed method?

**Questions:**

see Weaknesses

---

### Official Review · Reviewer_K4Cu · 2024-11-03

**Soundness:** 3
**Presentation:** 3
**Contribution:** 3
**Rating:** 8
**Confidence:** 3

**Summary:**

The paper proposes a novel Differential Pyramid Representation Network (DPRNet) designed for tone mapping high dynamic range (HDR) images to low dynamic range (LDR) outputs. DPRNet introduces a hybrid approach that combines global and local tone mapping modules with a Learnable Differential Pyramid (LDP) module, which helps capture multi-scale high-frequency details. The network’s architecture is organized into Global Tone Perception (GTP) for consistent luminance adjustments, Local Tone Tuning (LTT) for enhanced regional contrast, and Iterative Detail Enhancement for restoring finer details. The authors claim that DPRNet achieves significant improvements over state-of-the-art methods on benchmark datasets like HDR+ and HDRI Haven, with extensive experiments, visual comparisons, and ablation studies supporting their claims.

**Strengths:**

Innovation: The paper presents a novel solution that successfully integrates global and local tone mapping, tackling common HDR-to-LDR conversion challenges by balancing global consistency and local detail.

Method: DPRNet’s architecture, particularly the LDP and iterative enhancement modules, effectively preserves high-frequency details. This is a significant advantage over traditional tone mapping, which often struggles with fine textures.

Evaluation: The authors present robust quantitative improvements across various image quality metrics, including PSNR and SSIM. Qualitative comparisons and user studies further underscore DPRNet’s advantages in detail retention and aesthetic quality. Testing across multiple datasets (HDR+, HDRI Haven, HDR Survey, and UVTM video) demonstrates strong generalization, a critical factor for practical HDR applications.

**Weaknesses:**

Computational Complexity: While effective, DPRNet is computationally intensive, particularly with high-resolution images. The paper’s discussion of runtime and computational load is limited, which might impact its scalability for real-time or mobile applications.

Limited Benchmark Comparison: Although the authors provide comparisons with recent state-of-the-art methods, some relevant techniques are omitted, particularly those with similar high-frequency-focused approaches (e.g., S. Moran et al., ‘‘CURL: Neural curve layers for global image enhancement,’’ in ICPR, 2021 and L. Zhao et al., "Learning Tone Curves for Local Image Enhancement," in IEEE Access, 2022).

Generalization on Noisy Datasets: DPRNet struggles on datasets like MIT FiveK due to noise and overexposure, as noted by the authors. The paper could benefit from discussing potential solutions or additional noise-handling components.

User Study: While a user study is included, the scope could be broader to include users with different levels of experience in HDR processing, providing a more diverse evaluation of perceived quality.

**Questions:**

How would the method perform in real time? how expensive it would be to in terms of image resolution?

---

### Note · Authors · 2024-11-20

I have read and agree with the venue's withdrawal policy on behalf of myself and my co-authors.